# The Multi-Omics Landscape and Clinical Relevance of the Immunological Signature of Phagocytosis Regulators: Implications for Risk Classification and Frontline Therapies in Skin Cutaneous Melanoma

**DOI:** 10.3390/cancers14153582

**Published:** 2022-07-22

**Authors:** Jiahua Xing, Lingli Guo, Ziqi Jia, Yan Li, Yan Han

**Affiliations:** 1The First Medical Center, Department of Plastic and Reconstructive Surgery, Chinese PLA General Hospital, Beijing 100853, China; xingjiahuanku@163.com (J.X.); guo_linglidoctor@163.com (L.G.); ly527339819@163.com (Y.L.); 2School of Medicine, Nankai University, Tianjin 300071, China; 3Peking Union Medical College, Chinese Academy of Medical Sciences, Beijing 100730, China; ziqijia_19@student.pumc.edu.cn

**Keywords:** phagocytosis regulators, skin cutaneous melanoma, tumor-associated macrophages, tumor immune cell infiltration, subtype division

## Abstract

**Simple Summary:**

In this study, we focused on exploring phagocytosis regulators’ expression and mutational characteristics in skin cutaneous melanoma samples and delineating two molecular subtypes based on expression characteristics. We determined the relationship between phagocytosis regulators and survival by survival analysis of molecular subtypes. We then constructed a survival model (PRRS) to further quantify the criteria. Moreover, we combined pathway analysis, immune infiltration analysis, and mutation analysis to deeply explore the effects of phagocytosis regulators on skin cutaneous melanoma samples.

**Abstract:**

Tumor-associated macrophages (TAMs) have gained considerable attention as therapeutic targets. Monoclonal antibody treatments directed against tumor antigens contribute significantly to cancer cell clearance by activating macrophages to phagocytose tumor cells. Due to its complicated genetic and molecular pathways, skin cutaneous melanoma (SKCM) has not yet attained the expected clinical efficacy and prognosis when compared to other skin cancers. Therefore, we chose TAMs as an entrance point. This study aimed to thoroughly assess the dysregulation and regulatory role of phagocytosis regulators in SKCM, as well as to understand their regulatory patterns in SKCM. This study subtyped prognosis-related phagocytosis regulators to investigate prognostic differences between subtypes. Then, we screened prognostic factors and constructed phagocytosis-related scoring models for survival prediction using differentially expressed genes (DEGs) between subtypes. Additionally, we investigated alternative treatment options using chemotherapeutic drug response data and clinical cohort treatment data. We first characterized and generalized phagocytosis regulators in SKCM and extensively examined the tumor immune cell infiltration. We created two phagocytosis regulator-related system (PRRS) phenotypes and derived PRRS scores using a principal component analysis (PCA) technique. We discovered that subtypes with low PRRS scores had a poor prognosis and decreased immune checkpoint-associated gene expression levels. We observed significant therapeutic and clinical improvements in patients with higher PRRS scores. Our findings imply that the PRRS scoring system can be employed as an independent and robust prognostic biomarker, serving as a critical reference point for developing novel immunotherapeutic methods.

## 1. Introduction

Currently, we know that phagocytosis is involved in a variety of developmental, homeostatic, and tissue dynamic homeostasis processes inside the organism. It is crucial in tumor surveillance [1,2], foreign pathogen defense, neutralization, and eradication [3], as well as apoptosis, the clearance of cellular detritus after injury [4,5,6,7], and synaptic pruning [8]. Phagocytes ingest a variety of different particles via a variety of surface receptors and signaling cascades on their cell membranes [9]. Moreover, dysfunction in phagocytosis can result in immune system malfunction, aberrant protein aggregation, and developmental abnormalities [9,10].

In recent years, tumor-associated macrophages (TAMs) have gained increased attention as a potential target for tumor therapy. Macrophages, a substantial component of leukocyte infiltration, are present in varying amounts throughout all malignancies, and macrophages are essential inflammatory mediators in tumors [11]. TAM plays a critical function in cancer-related inflammation as a coordinator (CRI) [12]. It promotes tumor development on multiple levels, including promoting genetic instability, cultivating cancer stem cells, establishing metastasis, and suppressing adaptive protective immunity. Additionally, it exhibits T cell checkpoint activation triggers, and is thus a target for checkpoint blockade treatment [13].

Skin cutaneous melanoma (SKCM) is a malignant tumor arising from melanocytes of the skin and other organs, mainly in the skin and mucous membranes. Its pathogenesis is currently unknown, and extensive global data suggests that the most critical risk factor for the disease is overexposure to UV light [14]. The WHO has now designated indoor tanning as a carcinogen [15], and data from the United States in 2018 suggest that age, race, and gender are independent risk factors for SKCM [16]. The incidence of SKCM has climbed yearly, growing from 1 per 100,000 in 1940 to 17.74 per 100,000 in 2000 in the United States [17]. Furthermore, more than 100,000 cases of SKCM were reported in the United States in 2021 [18]. Its incidence is growing faster than all other cancer types, with approximately 55,000 deaths from SKCM worldwide yearly [19]. Its incidence in 2019 ranked third among men (684,470 cases) and fifth among women (672,140 cases) [20]. Compared with other skin tumors, SKCM has clinical features such as high late mortality, high recurrence rate, and high drug resistance due to its complex genetic and molecular mechanisms, making its clinical outcome and prognosis consistently unsatisfactory [21,22].

A study by Georgoudaki et al. revealed that TAM-mediated phagocytosis could influence tumor stem cell development and differentiation, tumor invasion and migration, and immune resistance. TAM can influence tumor cell growth and metastatic spread through immunosuppression in the tumor microenvironment (TME), thus serving as a critical target for tumor therapy [23]. Yamazaki et al.’s study suggested that phagocytosis can affect the TME as well, and thus interfere with tumor proliferation and differentiation, which has a promising future in tumor therapy [24]. TAM-mediated phagocytosis not only plays a role in the genesis of SKCM tumors, it promotes tumor drug resistance as well, and may exist as a constitutive state of metastatic SKCM cells [25]. The primary (M0 type) TAM can be differentiated into M1 and M2 types, which play different roles in the tumor, by recruitment and polarization of different cytokines in the TME [26]. The M1 type can inhibit tumor progression, and the M2 type can promote tumor growth [27]. The phenotype of TAM in SKCM is essential to analyzing tumor progression and identifying personalized therapies. According to Falleni et al., TAM has an M1 to M2 type switch during SKCM progression. Moreover, the accumulation of M2 type TAM is more significant than M1 type throughout the tumor development stage, thus favoring tumor growth and strongly correlating with poor prognosis [28]. In addition, Gordan et al. revealed that TAM can express PD-1 in humans and mice and that TAM PD-1 expression increases over time and with disease progression. TAM expression of PD-1 is negatively correlated with phagocytosis of antitumor cells; blocking PD-1/PD-L1 increases TAM phagocytosis and slows tumor growth. This suggests that immunotherapies such as PD-1/PD-L1 can operate directly on TAM, which has crucial implications for their use in tumor treatment [29]. Thus, TAM is a critical entry point for immunotherapeutic treatment of SKCM. Identifying and characterizing the regulation of TAM by phagocytosis regulators plays a crucial role in resolving the mechanism of cytophagocytosis in SKCM.

We employed a combination of several databases in this study to distill and summarize phagocytosis regulators discovered in prior research. We used the phagocytosis regulator-related system (PRRS) to develop a survival prediction model and assess patient prognosis by fully integrating immune cell infiltration within SKCM. Additionally, we estimated tumor micro-environment (TME) patterns in patients with high and low PRRS subgroups, and validated the PRRS scoring method using multiple model validation and clinical validation.

## 2. Material and Methods

### 2.1. Data Collection, Functional Analysis of Phagocytosis Regulators and Aberrant Phagocytosis in Tumors

First, we obtained relevant expression, phenotype, survival, gene copy, and gene mutation data from the TCGA database. The appropriate GEO data was downloaded from the GEO database. Following that, we downloaded the human gtf files from the Ensemble database. We obtained the set of immune cell-associated genes from Pornpimol Charoentong et al. [30]. We downloaded the relationship between immune infiltrating cells and gene expression from the TIMER database. We organized the data related to this study in Table 1, Table 2 and Table 3.

Afterward, we investigated how phagocytosis regulators affect macrophage phagocytosis. We collected phagocytosis regulatory factors for later analysis from Roarke A. Kamber’s [31] and Michael S. Haney’s [32] studies using a 5% false discovery rate (FDR) threshold. We combined them with those from Meghan A. Morrissey’s [33] study to access the complete phagocytosis regulators set. We calculated macrophage enrichment scores for cancer samples using the GSVA package in R and normalized the data using the scale function. We then calculated the Pearson correlation between macrophage enrichment scores and phagocytosis regulators’ expression, grouped the top six genes’ expression into high and low expression groups using the median as nodes, and examined the macrophages’ enrichment scores in the high and low expression groups. Additionally, we utilized the clusterProfiler package in R to conduct functional analysis on phagocytosis regulators, screening them against *p*-value < 0.05 and q-value < 0.2 to identify critical enrichment pathways.

Furthermore, we investigated abnormal phagocytosis in SKCM. The relevant gene sets were retrieved from the Molecular Signatures Database (https://www.gsea-msigdb.org/gsea/msigdb/index.jsp, accessed on 26 January 2022) using the keyword “phagocytosis”, which comprised 34 results; the samples were then grouped by age (>60 vs. ≤60) and stage (III–IV vs. I–II). Gene wet enrichment analysis (GSEA) was performed based on the software default thresholds, taking significant results (*p* < 0.05 and FDR < 0.25) for presentation. 

### 2.2. Genetic and Transcriptional Analysis

We started by investigating transcriptional and genetic changes in phagocytosis regulators in SKCM. We studied the differences in the expression of phagocytosis regulators between various subgroups of cancer. Then, we downloaded the MAF files of TCGA-SKCM mutations and mapped the mutation landscapes of phagocytosis regulators using the R package Map Tools. Next, we obtained the TCGA-SKCM data on gene-level copy number variation, counted the copy number changes of phagocytosis regulators, and estimated variation frequencies. We identified the positions of phagocytosis regulators from human chromosomal GTF data and utilized the R package RCircos to generate a gene circos map for position visualization.

The effect of phagocytosis regulators on tumor survival was then investigated. We combined overall survival (OS) data using univariate Cox regression analysis to identify genes associated with survival (*p* < 0.05), then divided the high and low expression subgroups based on the phagocytosis regulators significantly associated with survival, using their median as the node. To further investigate the correlation between genes, the Kaplan–Meier (KM) curves of gene expression groups were plotted using the R packages survival and survminer.

Furthermore, we extracted mutational information for phagocytosis regulators, screened genes that were mutated in at least five samples (1 percent of total samples), and screened genes that were significantly associated with survival based on mutation (*p* < 0.05) using univariate Cox regression. The KM curves were then plotted based on mutation with or without grouping in combination with OS for presentation.

### 2.3. Subtype Identification and Tumor Microenvironment Analysis

To begin, we classified molecular subtypes using phagocytosis regulators. We used the R package Consensus Cluster Plus to perform unsupervised clustering of samples based on the expression matrix of phagocytosis-regulated genes. The algorithm used was kmdist, and the distance was Euclidean. We then obtained molecular subtypes of phagocytosis-regulated genes using the PAC algorithm, followed by survival analysis of samples between subtypes (cluster 1 and cluster 2) and plotting of OS-based KM curves. At the same time, we performed principal component analysis (PCA) with the R packages factoextra and FactoMineR, then drew PCA scatter plots.

Following that, we performed TME engraving among several phagocytosis subtypes. Using the Hallmark gene set, we first performed GSEA of samples comparing subtypes of phagocytosis-regulated gene molecules. Thereafter, using the estimate package in R, we calculated the Stromal Score, Immune Score, ESTIMATE Score, and Tumor Purity of cancer samples in TCGA-SKCM data, then quantified the differences in sample-related scores between subtypes. In addition, the GSVA R package was utilized to calculate the enrichment scores of 28 immune-infiltrating cells in cancer samples, obtain the results, and normalize the data with the scale function. The proportion of immune cells in cancer samples was calculated using the R package CIBERSORT and the online TIMER database. We further collected immune checkpoint-related genes from the study of Xing Huang [34], then counted the expression differences of related genes among different subtypes in the TCGA-SKCM data.

Later, we performed validation of regulatory mechanisms among different phagocytosis subtypes. We began screening differentially expressed genes (DEGs) between subtypes (cluster 1 and cluster 2) using the R package limma with *p* < 0.05 and |logFC| > 1. We used the R package clusterProfiler to perform functional analysis on DEGs using criteria of *p*-value < 0.05 and q-value < 0.2 to identify significantly enriched pathways. Following that, we performed unsupervised clustering of samples based on the DEG expression matrix using the R package ConsensusClusterPlus, with pam as the algorithm and Euclidean as the distance. We used the PAC algorithm to generate molecular subtypes based on DEGs (DEG cluster 1 and DEG cluster 2), followed by survival analysis and sample plotting between subtypes using OS-based KM curves. We further examined the expression of phagocytosis-regulated genes among DEG molecular subtypes.

### 2.4. Construction and Validation of a Scoring System on Phagocytosis Regulators

First, we performed the construction of a phagocytosis regulator-related system (PRRS). We performed bulk univariate Cox regression analysis for DEG of cancer samples combined with OS data. After regression analysis, we screened genes significantly associated with OS (*p* < 0.05) for PCA analysis to obtain principal component 1 and principal component 2, and calculated the score for each sample based on the following formula:(1)$$\mathrm{PRRS}=\sum \left(PC1_{i}+PC2_{i}\right)$$
where *i* denotes the sample, *PC1* denotes principal component 1, and *PC2* denotes principal component 2. Principal components 1 and 2 (*PC1* and *PC2*) were chosen as signature scores. This approach down-weighted contributions from genes that did not track with other set members in favor of the score of the set that had the greatest block of genes with the strongest correlations (or anti-correlations). Then, to define the PRRS, we used a formula similar to earlier research [35,36].

To validate the model’s efficacy, we divided the PRRS of TCGA-SKCM samples into high and low PRRS groups using the median as the node, combined with OS data, plotted KM curves, and determined the difference between high and low PRRS groups to be significant at *p* < 0.05. We then used the sample PRRS scores as the model prediction results and combined them with survival data to calculate the area under the curve (AUC) of the model at one, three, and five year values and plotted the receiver operating characteristic (ROC) curves. Afterwards, we downloaded the GSE54467 data from the GEO database. PCA was conducted on the genes highly associated with OS based on TCGA data to generate *PC1* and *PC2*, and PRRS scores were then obtained to confirm the efficacy of our model once again.

Following that, we verified whether PRRS was associated with tumor prognosis. To determine whether the PRRS score grouping of TCGA data was an independent prognostic factor, we performed a univariate Cox regression analysis with other prognostic factors (age, gender, stage, cluster). In addition, we used multivariate Cox regression to examine the overall prognosis of the above five components (including PRRS score grouping) to demonstrate that the PRRS score factor was an independent prognostic factor. To further confirm that PRRS score grouping was an independent predictive predictor, we merged GSE54467 data by age, gender, and stage. Additionally, we examined the association of PRRS with other clinical features. Finally, we calculated the PRRS differences across subgroups and examined their significance. We carried out additional validation in two datasets, GSE19234 and GSE65904, to confirm the effectiveness of our PRRS.

### 2.5. Molecular Mechanisms and Somatic Alteration Analysis

To explore the relation between PRRS and cancer hallmarks, we first calculated the correlation between PRRS and hallmark enrichment scores and explored the functional differences between different PRRS groupings. We obtained the differential expression ranking of genes based on high and low PRRS groupings using the R package limma, followed by GSEA of differential genes using the gseKEGG and gseGO functions of the R package clusterProfiler (*p* < 0.05).

Next, we explored the TME of different PRRS subgroups. We further examined the difference in GSVA-calculated enrichment scores of immune infiltrating cells between high and low PRRS subgroups and demonstrated differences in the proportion of immune cells in CIBERSORT and TIMER-calculated cancer samples between high and low PRRS subgroups.

In addition, we explored the differences in genomic alterations across different PRRS subgroups. We used Masked Copy Number Segment data of TCGA-SKCM samples to analyze copy number variation (CNV) changes between high and low PRRS groupings online by GenePattern GISTIC 2.0 (https://cloud.genepattern.org/gp/pages/index.jsf, accessed on 6 February 2022). Then, we visualized them using the R package maftools. Moreover, we used the MAF files of TCGA-SKCM mutations in combination with the high and low PRRS groupings, then utilized the R package maftools to plot the mutation landscapes of the samples between the groupings.

### 2.6. Potential Immunotherapy Strategies Analysis

We performed chemotherapy drug resistance prediction using the R package pRRophetic to predict the response of samples to 138 drugs to obtain the predicted IC_50_ values. We then counted the differences in IC_50_ of samples in the high and low PRRS groups. After multiple corrections with Bonferroni methods, six drugs with significant differences (*p* < 0.05) were selected for presentation.

The IMvigor210 immunotherapy group from earlier studies with complete clinical and transcriptome data was included in our study [37]. The effectiveness of an anti-PD-L1 antibody in people with advanced uroepithelial cancer is the main focus of the IMvigor210 cohort. The IMvigor210 cohort has been extensively employed in cancer studies as a high-quality and comprehensive immunotherapy cohort to assess the prognostic value of immunotherapy in various kinds of tumor prediction models [38,39,40,41]. We downloaded IMvigor210 bladder cancer data using the R package IMvigor210CoreBiologies and performed an exploration of immunotherapy response. We then performed PCA based on genes significantly associated with OS attained from TCGA to obtain PC1 and PC2, which in turn yielded PRRS scores. We divided high and low PRRS score subgroups using the median as the node, plotted KM curves for high and low PRRS score subgroups, and counted the differences of PRRS scores of different response groups.

### 2.7. Validation of mRNA Expression Levels by qRT-PCR

We used the following cell lines to validate critical genes in SKCM and normal skin tissues: A375 human melanoma cell line (American Type Culture Collection, American), SK-MEL-28 cell line (National Infrastructure of Cell Line Resource, China), human immortalized keratin-forming cell line (Hacat), and human melanocyte line (PIG1) (Shanghai Guandao Biological Engineering Company, China). All cells were cultured in RPMI-1640 culture medium + 10% fetal bovine serum, with the surrounding environment maintained at 37 °C and 5% CO_2_. We further validated the collected specimens of thirty pairs of SKCM and normal skin tissues. All experimental components were approved by the Human Research Ethics Committee of the General Hospital of the Chinese People’s Liberation Army (Chinese PLA General Hospital), and all patients had signed an informed consent form. We employed qRT-PCR to determine the relative expression of nine essential genes [42]. All primers we used were synthesized by Huada Corporation (Beijing, China). 

### 2.8. Statistical Analysis

We used the Wilcoxon test for comparisons between two groups, the Kruskal–Wallis test for group comparisons among more than two groups, and the chi-square test for comparisons between proportions. We utilized Kaplan–Meier curves to plot survival curves for each subgroup and log-rank tests to assess whether differences were statistically significant. We employed univariate and multivariate Cox analyses to screen for independent predictors of OS. We used Spearman and Pearson correlation analysis to analyze the correlation coefficients and the Mann–Whitney test to compare the ssGSEA scores of different taxa of immune cells or immune pathways. All statistical analyses were performed with R version 4.1.1 and considered two-tailed *p* < 0.05 to be statistically significant.

## 3. Result

### 3.1. Phagocytosis Regulators Regulate Macrophage Phagocytosis in Tumor Development

Figure 1A depicts the flowchart of the study, whereas Figure 1B schematically depicts the evolution of TAM in the tumor microenvironment and its mechanism of action. We identified 271 phagocytosis regulatory factors in the linked literature [31,32,33], 260 of which included information on their expression. Then, we calculated the Pearson correlation coefficients and *p*-values for the 260 phagocytosis regulators and macrophage enrichment scores and used the top six to generate scatter plots. The correlations between the macrophages and the genes shown in the plots were more than 0.8 and *p* < 0.01 (Figure 2A–F). Based on their median expression levels, we divided the top six genes into high and low expression groups; our results indicated that the macrophage enrichment scores for all genes in the high expression group were significantly higher than those in the low expression group, which was consistent with the correlation results (Figure 2G–L).

We then performed functional enrichment analysis on genes controlled by phagocytosis. Among these, the examination of GO enrichment was subdivided into three sections: biological process (BP), cellular component (CC), and molecular function (MF). BP-enriched pathways are primarily immune response-activating cell surface receptor signaling pathways, CC-enriched pathways are primarily mitochondrial inner membrane pathways, MF-enriched pathways are primarily transcription coactivator activity pathways, and KEGG-enriched pathways are primarily thermogenesis pathways, among others (Figure 2M–P).

To further investigate aberrant phagocytosis in tumors, we examined the functional enrichment of age grouping (>60 vs. ≤60) and stage (III–IV vs. I–II) of cancer samples in the set of 34 phagocytosis-related genes. Our results showed that we enriched five gene sets in the >60 groupings for the age grouping, then enriched 29 gene sets in the ≤60 groupings; however, none reached significant levels. Meanwhile, for the stage grouping we enriched 22 gene sets in III–IV, which reached significant levels with three pathways (Figure 2Q–S).

### 3.2. Phagocytosis Regulator Genes and Transcriptional Alterations in Tumors

We evaluated differences in the expression of phagocytosis regulatory variables between age, gender, pathologic M, pathologic N, pathologic T, tumor stage, Breslow depth, and Clark level subgroups in cancer samples. The findings revealed that 41 genes were differentially expressed in the age grouping, 19 genes in the gender grouping, 6 genes in the pathologic M grouping, 25 genes in the pathologic N grouping, 109 genes in the pathologic T grouping, 104 genes in the tumor stage subgroup, 131 genes in the Breslow depth subgroup, and 63 genes in the Clark level subgroup (Figure 3A–H). However, because there are so many phagocytosis regulators, the significant results were tabulated, and the details are presented in Appendix A.

We mapped the mutation profiles of phagocytosis regulators based on the MAF files for TCGA-SKCM mutations. Due to the large number of genes, only the top 20 were displayed; our results indicated that the mutation rate of the top 20 phagocytosis regulating factors was 66.17% in SKCM samples, with ZDBF2 having the greatest mutation rate followed by HDAC9 (Figure 3I). We calculated the copy number changes of phagocytosis regulators using the TCGA-SKCM gene-level copy number data. As there were more genes, the top 30 were presented, with RRAGA having the highest frequency of deletion and PTEN having the second highest (Figure 3J). Additionally, we determined the frequency of copy number alterations in the top 30 phagocytosis regulators (Figure 3K) and plotted the locations of all phagocytosis regulators on each chromosome (Figure 3L).

We examined the influence of phagocytosis regulators’ expression on survival using OS data. Due to the large number of genes involved, we first used univariate Cox regression to identify those significantly associated with survival. Our results showed that the expression of 84 of 260 phagocytosis regulators was strongly related to prognosis, with HR > 1 for 13 genes and HR < 1 for the remaining 71 genes (Appendix A). Then, the nine genes with the lowest *p*-value were listed. The KM curves of these nine genes are presented in Figure 4A–I. We used OS data to investigate the influence of phagocytosis regulator mutations on survival. Considering the large number of genes, we first screened for genes associated with survival using univariate Cox regression. Our results showed that 126 genes had mutations in at least five samples, with seven genes having mutations associated with prognosis (HR > 1 in four genes and HR < 1 in the remaining three genes). Following that, we created the KM curves for these seven genes (Figure 4J–P).

### 3.3. Different Modes of Phagocytosis Regulation in Tumors

We performed unsupervised clustering on phagocytosis regulators’ expression matrix samples (Figure 5A–C) and identified two subtypes with significantly different KM curves (Figure 5D). Cluster 1 survival curves declined significantly faster than cluster 2 survival curves, and our PCA scatter plots reveal a clear distinction between the two subtypes (Figure 5E). Additionally, there were significant differences in the expression of phagocytosis regulators between the two subtypes (Figure 5F).

We then etched the TME between distinct phagocytosis subtypes. Using the MSigDB database, we first performed an enrichment analysis of hallmark pathways within samples between subtypes and discovered that 37 out of 50 pathways had significant differences, including HALLMARK TNFA SIGNALING VIA NFKB and HALLMARK TGF BETA SIGN ALING, which were all significantly different between the two subtypes (Figure 5G). Additionally, we generated the StromalScore, ImmuneScore, ESTIMATEScore, and TumorPurity scores for cancer samples and quantified the differences in sample-related scores between subtypes. The StromalScore, ImmuneScore, and ESTIMATEScore scores were considerably higher in cluster 2 samples than in cluster 1, although the TumorPurity scores were significantly higher in cluster 1 samples than in cluster 2 (Figure 6A–D). The enrichment scores of 28 immune infiltrating cells were then computed in cancer samples. Our results indicated that all immune infiltrating cells were considerably different between the two subtypes, with activated B cells, activated CD4^+^ T cells, and activated CD8^+^ T cells scoring significantly higher in cluster 2 than in cluster 1 (Figure 6E). Additionally, we determined the fraction of immune cells in cancer samples using the R package CIBERSORT and the online TIMER database. The corresponding box plots are provided in Appendix A. Moreover, we evaluated the expression of immunological checkpoints between subtypes. Our results revealed that 43 of 45 immune checkpoints have differential expression, with CD274, CTLA4, PDCD1, and LAG3 being significantly overexpressed (*p* < 0.0001) in cluster 2 (Figure 6F).

Afterward, we validated the regulatory mechanisms underlying distinct phagocytosis subtypes. Furthermore, we used all protein-coding genes to screen for DEGs between subtypes and then identified 661 DEGs based on threshold values. On DEGs, we performed functional enrichment analysis. The GO enrichment analysis was split into three parts: T cell activation was the primary focus of BP enrichment pathways; the external side of the plasma membrane was the primary focus of CC enrichment pathways; cytokine receptor binding was the primary focus of MF enrichment pathways; and cytokine–cytokine receptor interaction was the primary focus of KEGG enrichment pathways (Figure 7A–D). We further performed unsupervised clustering on the samples using the DEG expression matrix. We obtained two subtypes with significant differences in the KM curves of the samples between the subtypes and significant differences in the expression of the DEGs between the two subtypes (Figure 7E–F). The expression of phagocytosis regulators was then compared between differential subtypes; our results indicate that the expression of several phagocytosis regulators varied between differential gene expression subtypes (Figure 7G).

### 3.4. Phagocytosis Regulator-Related System (PRRS)

We started by constructing the PRRS. We screened 582 survival-related DEGs using univariate Cox regression based on DEGs in TCGA data, then performed PCA on these 582 genes to obtain PC1 and PC2 for summation, attained PRRS scores, and then divided them into high and low PRRS groups with the median (the high and low PRRS groups are the different groups with PRRS scores above and below the median). The result showed that the difference in KM curves between the two groups was significant. Then, we used the sample risk score as the model prediction result and combined it with the survival data to calculate the AUC of the model. The AUCs at one, three, and five years were greater than 0.6, indicating good model efficacy (Figure 8A–E). We validated the model’s efficacy in GSE54467. In addition, we performed PCA based on the expression matrix of survival-related DEGs in GSE54467, attaining PC1 and PC2 and calculating their sum to obtain the PRRS. The median was used as the division to develop the high and low PRRS groups. The KM curves of the two groups differed substantially, and the AUC values calculated from the time ROC curves were all greater than 0.6 (Figure 8F–J). To further validate the efficacy of our PRRS, we conducted further validation in two datasets, GSE19234 and GSE65904, and both results showed statistically significant differences (*p* < 0.05), demonstrating the excellent performance of PRRS (Appendix A).

We further examined the independent prognostic effect of the PRRS subgroups using TCGA data. Our results revealed that PRRS subgroup, age subgroup, and stage subgroup all had greater prognostic efficacy (Figure 9A,B). Similarly, we combined age, gender, and stage to verify whether PRRS score subgroup was an independent prognostic factor in the GSE54467 data. Our results demonstrated that the PRRS score subgroups had better prognostic efficacy (Figure 9C,D).

Moreover, we counted the PRRS differences in age, gender, pathologic M, pathologic N, pathologic T, tumor stage, Breslow depth, Clark level, and molecular subtype groups. Our results showed that the PRRS score of pathologic T decreased gradually with increasing grade; the PRRS score of patients with Clark levels IV–V was significantly lower, the PRRS score of patients with tumor stage II was significantly lower than that of stage I, and the PRRS score of patients with cluster 2 in subtype classification was higher (Figure 9E).

### 3.5. Analysis of the Molecular Mechanisms of Different PRRS

We further examined the relationships between PRRS and cancer hallmark enrichment scores. Our results showed that most pathways were significantly correlated with PRRS, particularly HALLMARK TNFA SIGNALING VIA NFKB and HALLMARK APOPTOSIS, which were significantly positively correlated with PRRS, while HALLMARK DNA REPAIR and HALLMARK MYC TARGETS V2 were significantly negatively correlated with PRRS (Figure 10A).

Afterwards, we performed further GSEA-based GO and KEGG enrichment analysis based on PRRS grouping. For the GO enrichment results, the main enrichment pathways were activation of the innate and adaptive immune responses. Additionally, for the KEGG enrichment results the main enrichment pathways included cytokine–cytokine receptor interaction and endocytosis (Figure 10B,C).

Following that, we recorded the TME for each PRRS grouping. Our results showed that samples with a high PRRS had substantially higher StromalScore, ImmuneScore, and ESTIMATEScore than those with a low PRRS. However, for the TumorPurity score the results showed the inverse (Figure 10D–G). In addition, we counted the enrichment scores of 28 immune infiltrating cells between different PRRS groups. Our results indicated a statistically significant difference in the number of immune-infiltrating cells in the different PRRS groups. We concluded that all immune-infiltrating cell enrichment scores in high PRRS samples were considerably higher than in low PRRS samples (Figure 10H). Additionally, we evaluated the differences in immune-infiltrating cell proportions between high and low PRRS subgroups using CIBERSORT and online TIMER data. The corresponding box plots are depicted in Appendix A.

Furthermore, we investigated the differences in genomic changes between PRRS subgroups. The frequency of CNV changes was calculated between PRRS subgroups; our results indicate that CNV changes were much lower in high PRRS samples than in low PRRS samples (Figure 10I–K). Additionally, we counted the differences in CNV between genes in the PRRS subgroups and found 5202 genes with substantial CNV differences between the two groups. We investigated gene mutations in different PRRS groupings and discovered that the mutation rate was slightly higher in high PRRS samples than in low PRRS samples (Figure 10L,M). We further counted the genes that had significantly different mutations between the two groups of samples. Our findings indicated that 325 genes had significantly different mutations between the two groups.

### 3.6. Potential Treatment Strategies Based on PRRS

We predicted the reactions to 138 medications of samples in the high and low PRRS groups. We then used box plots to demonstrate the six with both substantial differences and common medications. Our results indicate that the high PRRS samples were more medication-resistant (Appendix A).

In addition, we collected IMvigor210 bladder cancer data and generated sample PRRS scores based on gene expression data within the model. The KM curves indicated no significant differences between the high and low PRRS groups. However, PRRS scores were considerably higher in complete response (CR) samples than in partial response (PR) or progressive disease (PD) samples, implying that PRRS scores may be related to immunotherapy response (Appendix A). Appendix A summarizes the endogenous processes of SKCM and their impact on the antitumor immune response.

### 3.7. Validation of Expression-Based Regulators of Survival Significantly Associated with Phagocytosis

We verified the expression levels of these nine highly associated phagocytosis regulators using qRT-PCR in human tissues and cell lines. Appendix A display the primer sequences and basic information about the patient specimens we used. Our results showed that ACTR3, AXL, CIITA, DOCK2, FCGR2A, FCRL3 were up-regulated in SKCM (Appendix A), as well as that BIN2, CD38, and FCGR1B were down-regulated in SKCM (Appendix A).

## 4. Discussion

SKCM is one of the most immunogenic tumors because of its high tumor mutation burden (TMB). Therefore, it has a high potential to respond to immunotherapy [43]. However, SKCM often remains resistant to immunotherapy, and the five-year survival rate for stage IV SKCM remains below 19% [44]. Immune cell infiltration (ICI) has recently been established as a robust and predictive biomarker of prognostic value in several types of malignancies, including SKCM [45,46,47]. Despite many clinical trial results highlighting the efficacy of immunotherapy in advanced metastatic melanoma, SKCM patients exhibit substantial heterogeneity in their response to immunotherapy. 35–60% of patients show a resistant response to PD-1 blocking immunotherapy [48], 40–65% exhibit mild resistance, and 43% develop acquired resistance [49]. In recent years, ICI have propelled SKCM to a new level of oncology treatment. Numerous studies have confirmed that the immunobiological components within the TME play a crucial role in immunotherapy response rates, patient prognosis, and tumor progression [50]. Therefore, a better elucidation of the TME immunophenotype can help reveal the biological mechanisms of tumor development. Previously, SKCM was divided into several immunological subtypes based on various levels of research. A study by Liu et al. constructed ICI scores based on PCA of immune signature genes of DEGs and identified three ICI gene clusters associated with different immune subtypes and survival outcomes. Their results showed that high ICI scores exhibited an activated immune profile and better prognosis [51]. Meanwhile, Hu et al.’s study used hierarchical clustering analysis to macroscopically classify SKCM into two stable subtypes, namely, the high immune and low immune groups, based on the heterogeneity of immune infiltration in patients. The high immune group expressed more HLA genes, robust immunogenicity, and longer survival time [52]. The CIBERSORT and ESTIMATE algorithms were employed in Zhao et al.’s work together with the TCGA and GEO datasets to define two immunophenotypes of uveal melanoma (UVM). UVM is considered an immune-cold tumor due to its low TMB (non-synonymous variants) and unique TME. The high ICI subtype has a poor prognosis [53]. A study by Chen et al. classified SKCM into four subtypes using epigenetic DNA methylation-correlated (METcor) genes and CNV-correlated (CNVcor) genes as the core. Immune cell scores were markedly elevated in the iC1 subtype, which had the best prognosis. The iC3 subtype, associated with the most aggressive SKCM cases, exhibited immune cell infiltration and significantly lower scores [54]. Our study is the first to focus on TAMs to delineate immune subtypes by phagocytosis regulators and construct a PRRS score based on TME to assess patient prognosis and the benefit of immunotherapy.

This study aimed to determine the function of phagocytosis regulators and their effect on the development of SKCM. We discovered that phagocytosis regulators can influence the number of phagocytes, while their expression or mutation can alter prognosis. We subsequently subtyped the samples based on prognosis-related phagocytosis regulators, discovered disparities in prognosis between subtypes, and established a correlation between phagocytosis regulators and survival through survival analysis of molecular subtypes. After that, we identified DEGs between subtypes, performed differential gene subtype identification and functional enrichment analysis, and looked for other differences between phagocytosis-related subtypes. Using DEGs between subtypes, we screened for prognostic factors to create a survival model. We established PRRS, a scoring system for predicting survival in patients. We validated the model’s efficacy numerous times and coupled pathway analysis, immune infiltration analysis, and mutation analysis to investigate the effects of phagocytosis regulators on SKCM samples. In addition, we probed potential treatment options using chemotherapy drug response and clinical cohort treatment response and described the model by linking it to known clinical characteristics. Moreover, we used qRT-PCR to re-validate expression-based regulators of survival significantly associated with phagocytosis. Our findings suggest the following: (i) phagocytosis regulators can be used to assess PRRS patterns in SKCM patients; (ii) there is a correlation between low PRRS scores and poor prognosis; (iii) there is a correlation between gene mutations and prognosis in SKCM, and (iv) there is a correlation between PRRS scores and ICB treatment response.

Numerous previous studies have confirmed the presence of three main immune phenotypes in tumors, namely, the immune-desert, immune-excluded, and immune-inflamed phenotypes [55]. We first applied unsupervised clustering to define two immune subtypes based on the phagocytosis regulators’ expression matrix. Cluster 1 had an immune-desert phenotype, with low levels of immune infiltrating cells in the tumor parenchyma and stroma as well as low immunological and stromal scores [56]. Cluster 2 exhibited an immune-inflamed phenotype, with high levels of CD4^+^ T cells and CD8^+^ T cells in the tumor parenchyma. This phenotype tends to have high expression of immune checkpoints and benefits more from immunotherapy [57]. TILs such as CD4^+^ T cells and CD8^+^ T cells play an essential role in tumor metastasis, recurrence, and response rate to immunotherapy. [58,59]. CD8^+^ T cells perform tumor-killing functions based on cell differentiation and infiltration [60]. Naive CD8^+^ T cells initiate a differentiation program after infiltration and further differentiate into cytotoxic and effector CD8^+^ T cells for anti-tumor functions [61]. CD4^+^ T cells have long been recognized as an important component of tumor immunotherapy, as they are capable of promoting or suppressing anti-tumor cytotoxic CD8^+^ T cell responses in secondary lymphoid organs or tumors, hence modulating the tumor immune microenvironment [62]. Cluster 1 resembles tertiary lymphoid structures and is found in the margins and interstitium of aggressive tumors, which are sites of immune cell activation and recruitment [55]. In addition, the number of TILs in cluster 1 TME is significantly less than in cluster 2. These features may explain the high expression of the tumor-associated Hallmark pathway in cluster 1, the low expression of immune checkpoints, and the overall poor prognosis. Our study showed that cluster 2 is correlated with better overall survival as an immune-inflamed phenotype and that its immune checkpoint expression is higher. We discovered that high levels of ICIs are predictive of better prognosis, which is consistent with previous research [63]. Next, we screened for DEGs among cluster subtypes and applied unsupervised clustering to define two DEG subtypes based on the DEG expression matrix. Similar to the immune subtype results, the DEG subtypes with a better prognosis and longer survival exhibited characteristics similar to cluster 2. These DEGs could potentially be independent and robust biomarkers in the future.

In light of the individual heterogeneity of the TME, it is essential to quantify the pattern of immune cell infiltration in individual tumors. Considering the correlation between SKCM immune subtypes and survival, we constructed PRRS scores to assess patient clinical characteristics, survival prognosis, and response to immunotherapy. As a “hot” tumor phenotype, the high PRRS score group showed high immunoreactivity and mesenchymal activity, significantly enriched immune-related pathways, and a better survival prognosis. In addition, the high PRRS group showed a high TMB. Tumor cells with high TMB have higher levels of neoantigens, which help the immune system to recognize tumors, thereby stimulating anti-tumor T cell proliferation and anti-tumor responses. Theoretically, the higher the TMB, the more neoantigen production can be recognized by T cells and the better the immunotherapeutic effect [64]. According to Yan et al., the lower TMB in SKCM patients results in poor survival prognosis and is correlated with an advanced pathological stage [65]. Our results are identical to previous studies, with a poorer prognosis in the low TMB group. Furthermore, the high PRRS group showed a lower frequency of CNV changes. Previous research has linked a high CNV change frequency to the development of a variety of complex diseases, such as Alzheimer’s disease [66], psoriasis [67], and congenital generalized hypertrichosis terminalis (CGHT) [68]. Therefore, we hypothesize that a high frequency of CNV changes in SKCM patients would lead to a poorer prognosis. However, more clinical data are needed to support the current conclusions. Based on our comprehensive analysis of SKCM phagocytosis regulators’ mutation profiles, we found that ZDBF2 has the highest mutation rate. Its mutation and methylation are associated with various diseases, including multilocus imprinting disturbances such as Temple syndrome (TS14) and Kagami–Ogata syndrome (KOS14) [69]. In contrast, an association between its deletion and Nasopalpebral lipoma-coloboma syndrome, a rare malformation, has been reported [70]. RRAGA (Rag A GTPase) is the most frequent deletion. Its mutations can affect lysosomal function and relocalization, autophagy, altered cell growth, and promoter activity. Mutations in RRAGA as a critical regulator of mTORC1 (mechanistic rapamycin complex 1) are closely associated with autosomal dominant cataracts [71]. These genes may be a breakthrough point for future SKCM immunotherapy.

Our findings suggest that PRRS scores can be used as independent and robust biomarkers to predict prognosis in SKCM patients. Ascierto et al.’s study revealed that higher levels of TILs such as CD8^+^ T cells and CD4^+^ T cells in the TME contribute to improved disease-free survival (DFS) with immunotherapy [72]. We evaluated IMvigor210 data in patients receiving immunotherapy and found that PRRS scores were significantly higher in effective clinical remission with immunotherapy patients. This indicates that the high PRRS group has an activated immune system and may benefit more from immunotherapy, which is consistent with previous studies. By analyzing the main clinical characteristics, patients with high PRRS scores had a better prognosis than those with low scores. The PRRS score accurately predicted the prognosis of patients in the pathological T-stage, tumor stage, and Clark stage. We discovered that cluster 2, an immune-inflamed phenotype with a higher PRRS score, was more likely to benefit from immunotherapy, proving the accuracy of our scoring system once more.

## 5. Conclusions

In summary, this study has elucidated the dysregulation and regulatory significance of phagocytosis regulators in SKCM, established their relationship with tumor heterogeneity, and examined their regulatory patterns in SKCM. This study provides a comprehensive investigation of multiple databases, pointing the way forward in terms of the tumor microenvironment, functional analysis, immune response, and drug sensitivity, and has significant implications for the development of novel immune drugs and the promotion of personalized immunotherapeutic strategies.

## Figures and Tables

**Figure 1 cancers-14-03582-f001:**
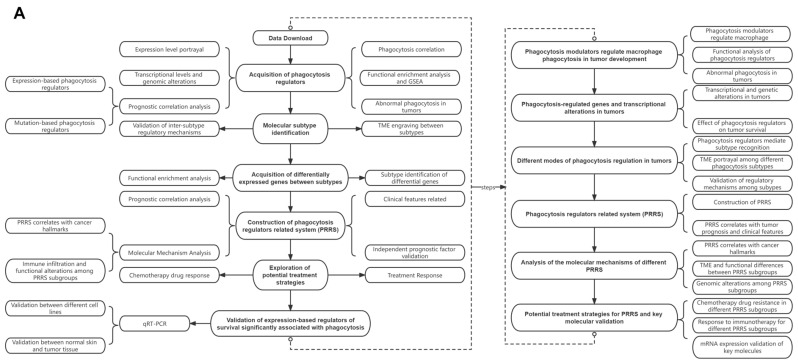
(**A**) The flowchart of this study and (**B**) a schematic diagram of the role of tumor-associated phagocytes (TAMs). TAMs are infiltrating tumor tissue macrophages, and are mainly differentiated from monocytes. Chemokines such as CSF1 and CCL2 secreted by tumor cells can recruit monocytes from peripheral circulating blood to the tumor microenvironment (TME), after which monocytes differentiate into macrophages. TAMs are an essential component of the TME, and are often associated with poor prognosis and drug resistance; they have emerged as very promising targets in cancer immunotherapy.

**Figure 2 cancers-14-03582-f002:**
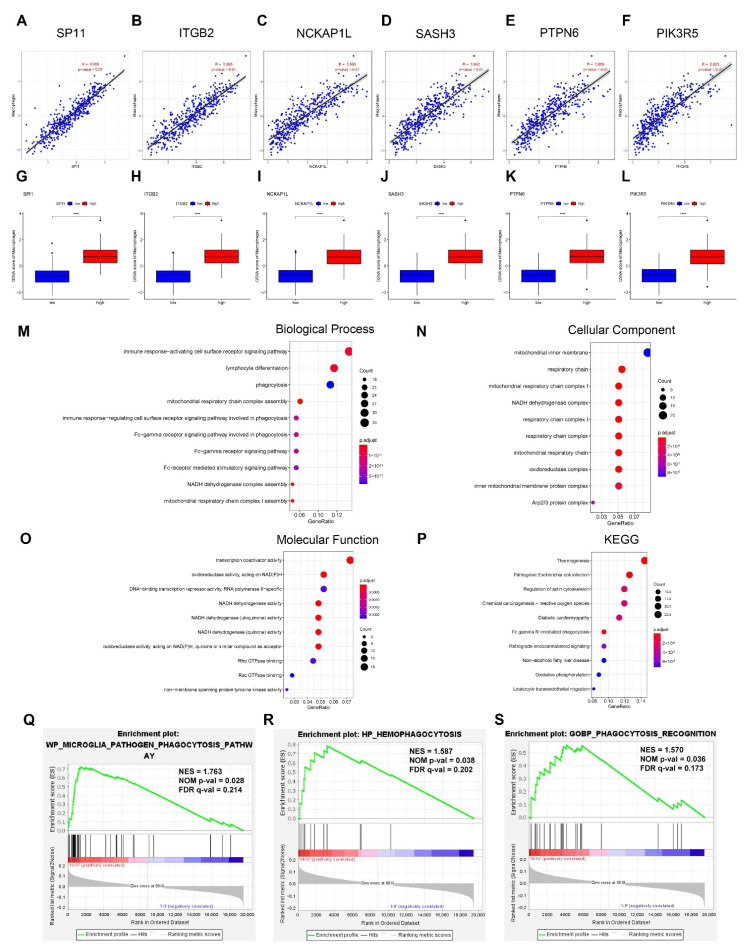
Phagocytosis regulators regulate the involvement of macrophages in tumor development. (**A**–**L**) Phagocytosis regulators regulate macrophage phagocytosis. (**A**–**F**) Pearson correlation of phagocytosis regulators with macrophage enrichment scores (Top6). We obtained 260 phagocytosis regulators with expression information from the relevant literature and then calculated their Pearson correlations with macrophage enrichment scores. We took the six with the strongest correlations (**A**–**F**: SP11, ITGB2, NCKAP1L, SASH3, PTPN6, PIK3R5) to plot scatter plots (all *p*-values < 0.01 and correlation coefficients > 0.8). (**G**–**L**) Differences in macrophage enrichment scores in high and low gene expression groups. We divided the top six genes among the 260 phagocytosis regulators into high and low expression groups according to the median. Our results showed that the macrophage enrichment scores were significantly higher in the high expression group than in the low expression group for all genes (**** *p* < 0.0001). (**M**–**P**) Functional analysis of phagocytosis regulators. (**M**) Functional analysis of biological process. (**N**) Functional analysis of cellular component. (**O**) Functional analysis of molecular function. (**P**) Functional analysis of KEGG. (**Q**–**S**) Abnormal phagocytosis in tumors. We examined the functional enrichment of stage groupings of cancer samples in a set of 34 phagocytosis-associated genes. The following three pathways reached significant enrichment levels (*p* < 0.01): (**Q**) WP microglia pathogen phagocytosis pathway; (**R**) HP hemophagocytosis pathway; And (**S**) GOBP phagocytosis recognition pathway.

**Figure 3 cancers-14-03582-f003:**
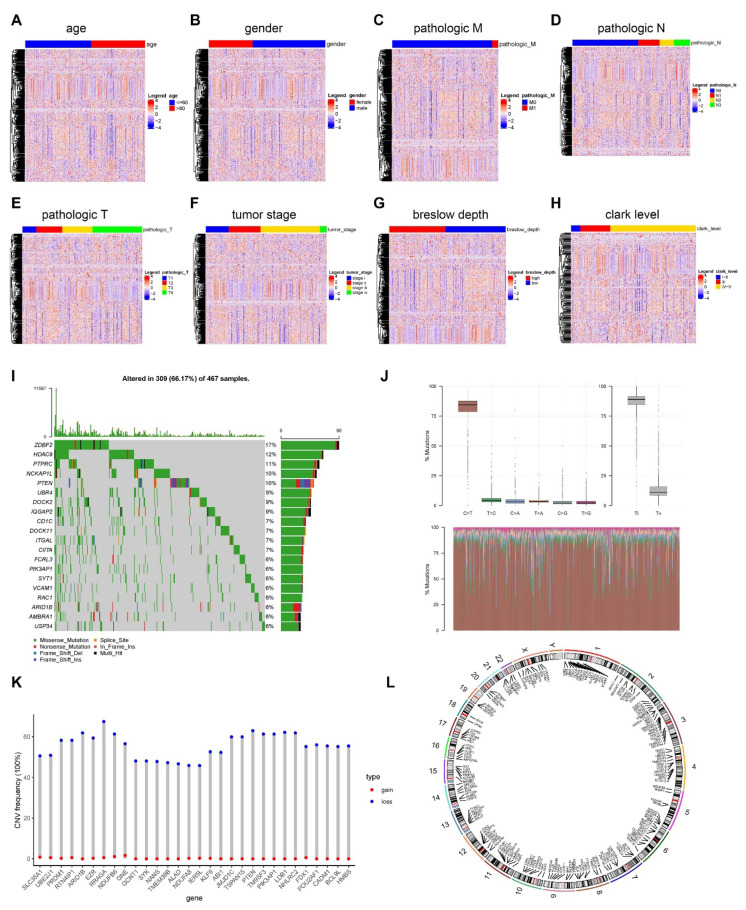
Transcriptional and genetic alterations of phagocytosis regulators in tumors. (**A**–**H**) Differential expression of phagocytosis regulatory factors in cancer samples across clinical factors, including age (41 genes were differentially expressed), gender (19 genes were differentially expressed), pathologic M (6 genes were differentially expressed), pathologic N (25 genes were differentially expressed), pathologic T (109 genes were differentially expressed), tumor stage (104 genes were differentially expressed), Breslow depth (131 genes were differentially expressed), Clark level (63 genes were differentially expressed). (**I**–**L**) Genomic alterations in phagocytosis regulators. (**I**) Mutation profile of the top 20 phagocytosis regulators. Our results showed that mutations occurred in 309 of the 467 samples, accounting for 66.17% of the samples. (**J**) Sample mutation frequency in 467 skin cutaneous melanoma samples. (**K**) Copy number change frequency statistics of the top 30 phagocytosis regulators. Our results demonstrated that the highest frequency of deletion occurred in RRAGA. (**L**) The positions of all phagocytosis regulators on each chromosome. The connector (small line segment connecting gene names) marks the gene’s location on the chromosome. Connectors that are more dense indicate a higher level of enrichment. We infer that the regions on chromosomes 1 and 3 are more enriched than other chromosome regions.

**Figure 4 cancers-14-03582-f004:**
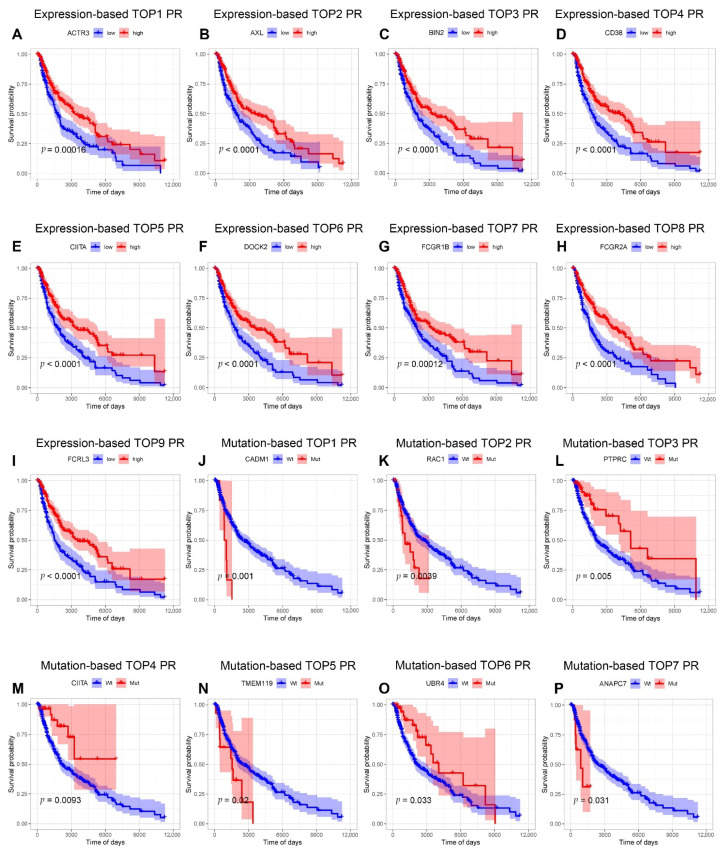
Effect of phagocytosis regulators on tumor survival. (**A**–**I**) We investigated the effect of phagocytosis regulators’ expression on the survival of skin cutaneous melanoma patients. The expression of 84 genes out of 260 phagocytosis regulators was significantly associated with prognosis. We used the nine with the lowest *p* values to plot Kaplan-Meier curves, including ACTR3, AXL, BIN2, CD38, CIITA, DOCK2, FCGR1B, FCGR2A, and FCRL3 (*p* < 0.0001). (**J**–**P**) We investigated the effect of mutations in phagocytosis regulators on the survival of skin cutaneous melanoma patients. Among the 260 phagocytosis regulators, 126 genes met the requirement of being mutated in at least five samples, and seven of these genes were significantly associated with prognosis: CADM1, RAC1, PTPRC, CIITA, TMEM119, UBR4 and ANAPC7.

**Figure 5 cancers-14-03582-f005:**
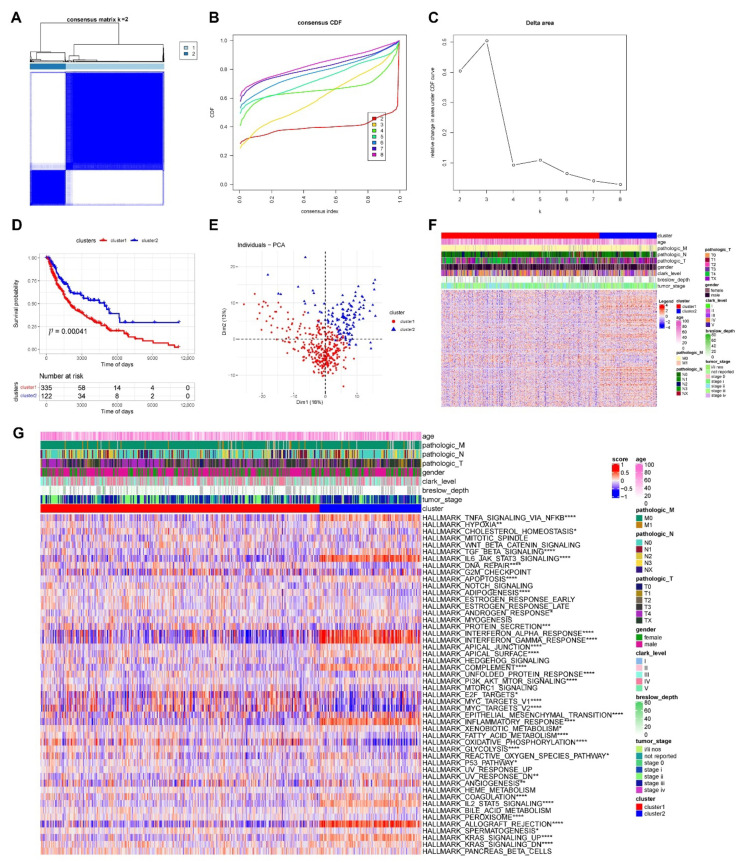
Different modes of phagocytosis regulation in tumors. (**A**–**F**) Identification of molecular subtypes of phagocytosis regulators. (**A**) Heat map of unsupervised clustering of samples. (**B**) Cumulative distribution graph. (**C**) Unsupervised clustering gravel map. (**D**) Inter-subtype Overall Survival (OS)-based Kaplan-Meier curves. (**E**) Principal component analysis (PCA) scatter plot. (**F**) Heat map of phagocytosis regulators expression. (**G**) Heat map display of enrichment analysis results of the inter-subtype hallmark pathway. We performed MSigDB database-based hallmark pathway enrichment analysis on samples between subtypes, with 37 of the 50 pathways showing significant differences (* *p* < 0.05, ** *p* < 0.01, *** *p* < 0.001, **** *p* < 0.0001).

**Figure 6 cancers-14-03582-f006:**
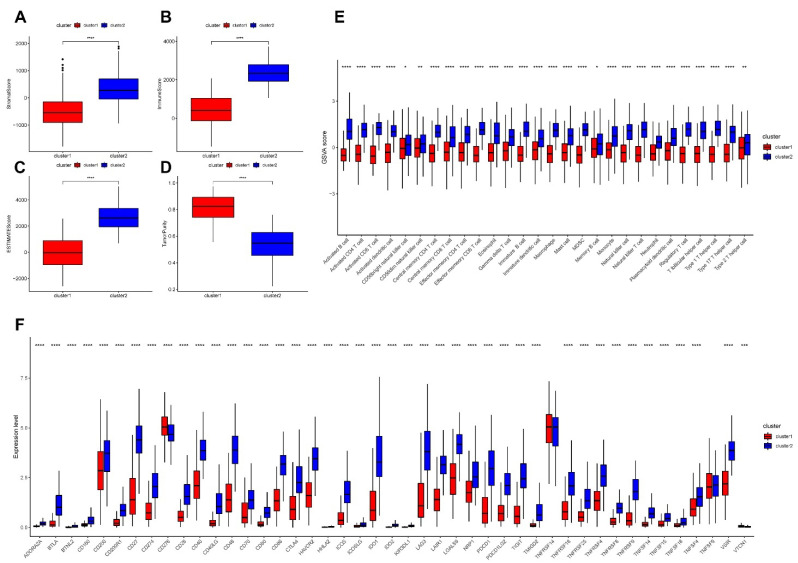
(**A**–**E**) Differences in immune infiltrating cells between subtypes. (**A**) Immunization score difference display of StromalScore. (**B**) Immunization score difference display of ImmuneScore. (**C**) Immunization score difference display of ESTIMATEScore. (**D**) Immunization score difference display of TumorPurity. (**E**) Differences in immuno-infiltrating cell enrichment scores (* *p* < 0.05, ** *p* < 0.01, **** *p* < 0.0001). (**F**) Differences in immune checkpoint expression between subtypes. Our results showed significant expression differences in 43 of the 45 immune checkpoints (*** *p* < 0.001, **** *p* < 0.0001).

**Figure 7 cancers-14-03582-f007:**
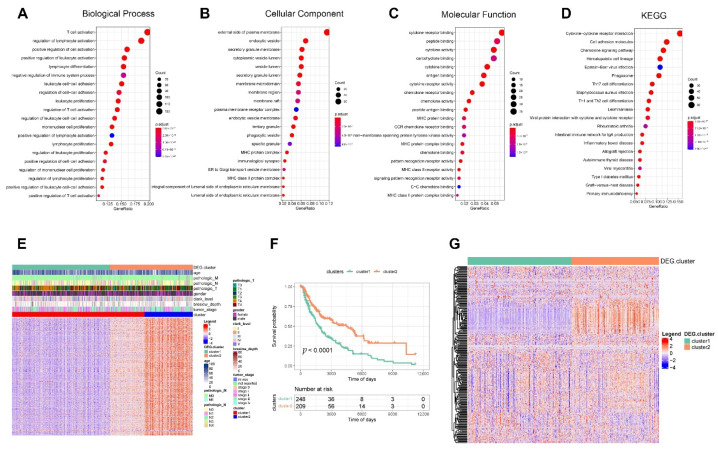
Validation of regulatory mechanisms among different phagocytosis subtypes. (**A**–**D**) Functional enrichment analysis of differentially expressed genes (DEGs). We screened for DEGs between subtypes based on protein-coding genes, obtained 661 DEGs based on thresholds, and then performed functional enrichment analysis on DEGs. (**A**) Functional analysis of biological process. (**B**) Functional analysis of cellular component. (**C**) Functional analysis of molecular function. (**D**) Functional analysis of KEGG. (**E**,**F**) Unsupervised clustering based on DEGs. (**E**) Expression heat map of DEGs. (**F**) Kaplan-Meier curves between differential genetic subtypes. Among them, cluster 1 contains 248 samples and cluster 2 contains 209 samples. (**G**) Heat map of the expression of phagocytosis regulators among differential genetic subtypes. Our results showed significant differences in the expression of many phagocytosis regulators between subtypes.

**Figure 8 cancers-14-03582-f008:**
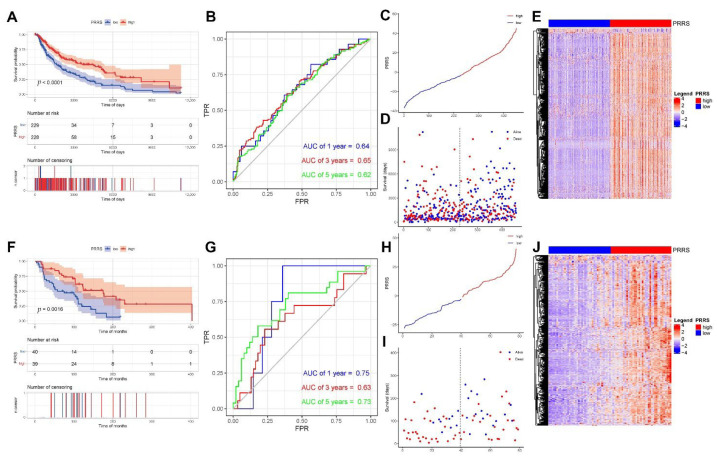
Phagocytosis regulator-related system (PRRS) construction. (**A**–**E**) Model effectiveness validation of PRRS for TCGA data. We performed univariate Cox regression based on DEGs in TCGA to screen 582 DEGs associated with survival and then performed PCA to obtain PRRS scores based on these genes. (**A**) Build model Kaplan-Meier curve validation. (**B**) Receiver operating characteristic (ROC) curve validation. The areas under the curve (AUC) were all greater than 0.6, indicating good model efficacy. (**C**) PRRS score graphs for all samples. (**D**) Scatter plot of survival time for all samples. (**E**) Heat map of gene expression in the model. (**F**–**J**) Model effectiveness validation of PRRS for GSE54467 data. (**F**) Build model Kaplan–Meier curve validation. (**G**) ROC curve validation. The AUCs were all greater than 0.6, indicating good model efficacy. (**H**) PRRS score graphs for all samples. (**I**) Scatter plot of survival time for all samples. (**J**) Heat map of gene expression in the model.

**Figure 9 cancers-14-03582-f009:**
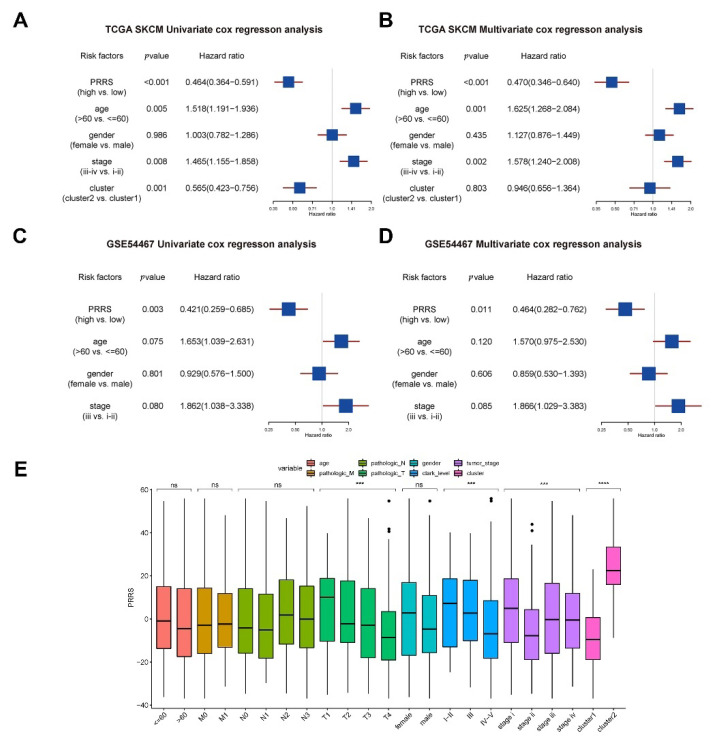
PRRS correlates with tumor prognosis and clinical features. (**A**,**B**) We used TCGA-SKCM data to verify whether PRRS grouping is an independent prognostic factor. Our results demonstrated that PRRS subgroup (*p* < 0.001), age subgroup (*p* = 0.005) and stage subgroup (*p* = 0.008) had better prognostic efficacy and were independent of each other. (**A**) TCGA-SKCM univariate cox regression analysis. (**B**) TCGA-SKCM multivariate cox regression analysis. (**C**,**D**) We used GSE54467 data to verify whether PRRS grouping is an independent prognostic factor. Our results demonstrated a better prognostic efficacy for PRRS subgroup grouping (*p* = 0.003). (**C**) GSE54467 univariate cox regression analysis. (**D**) GSE54467 multivariate cox regression analysis. (**E**) Differences in PRRS in clinical characteristic groupings (ns, not significant, *** *p* < 0.001, **** *p* < 0.0001).

**Figure 10 cancers-14-03582-f010:**
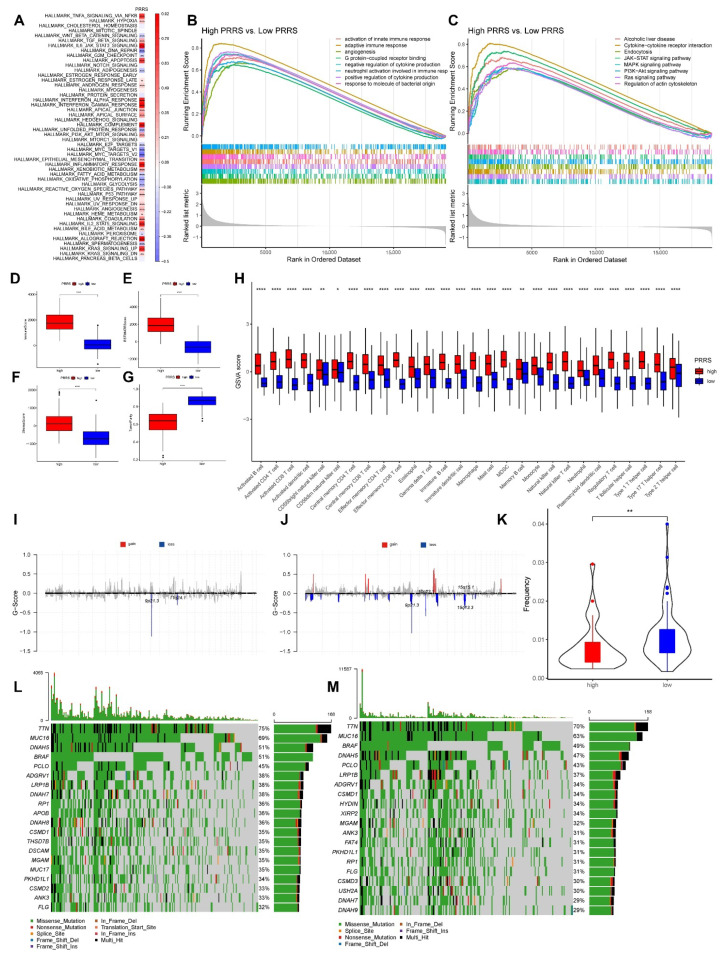
Analysis of the molecular mechanisms of different PRRS. (**A**) Correlation display of PRRS and hallmarks. Our results demonstrated that most of the pathways were significantly correlated with PRRS (* *p* < 0.05, ** *p* < 0.01, *** *p* < 0.001). (**B**,**C**) Functional differences between PRRS subgroups. (**B**) Display of GO function enrichment results for PRRS grouping. (**C**) Display of KEGG function enrichment results for PRRS grouping. (**D**–**H**) Differences in immune infiltrating cells between high and low PRRS subgroups (* *p* < 0.05, ** *p* < 0.01, **** *p* < 0.0001). (**D**) Differences in ImmuneScore in subgroups. (**E**) Differences in ESTIMATEScore in subgroups. (**F**) Differences in StromalScore in subgroups. (**G**) Differences in TumorPurity in subgroups. (**H**) Differences in immune-infiltrating cell enrichment scores between high and low PRRS subgroups were demonstrated. Our results showed significant differences in all immune-infiltrating cells in the PRRS grouping. (**I**–**K**) Gene copy number variations (CNV) profiles among PRRS subgroups. Our results showed that the frequency of CNV changes was significantly lower in high PRRS samples than in low PRRS samples, and there were 5202 genes with significant differences in CNV between the two groups (*p* < 0.001). (**I**) CNV profile in high PRRS samples. (**J**) CNV profile in low PRRS samples. (**K**) CNV frequency statistics between the two groups. (**L**,**M**) Gene mutations among PRRS subgroups. Our results showed a slightly higher mutation rate in high PRRS samples than in low PRRS samples, with 325 genes significantly different between the two groups (*p* < 0.001). (**L**) Mutation of high PRRS samples. (**M**) Mutation of low PRRS samples.

**Table 1 cancers-14-03582-t001:** Characteristics of patients in TCGA-SKCM.

Variables	Number of Cases
***Age*****(*years*)**≤60/>60	251/206
***Gender***Male/Female	285/172
***Stage***0/I/II/III/IV/I or II nos/not reported	6/77/136/170/23/10/35
***Pathologic T***T_0_/T_1_/T_2_/T_3_/T_4_/T_x_	23/41/77/90/148/78
***Pathologic N***N_0_/N_1_/N_2_/N_3_/N_x_/NA	226/73/49/56/34/19
***Pathologic M***M_0_/M_1_/NA	407/24/26
***OS***Alive/Dead	235/222
***Cluster*^1^**cluster 1/cluster 2***DEG cluster*^2^**DEG cluster 1/DEG cluster 2***Clark level***I/II/III/IV/V***Breslow depth***>median(high)/≤median(low)/NA	
335/122

248/209

5/18/76/164/52

170/182/105

^1^ molecular subtypes; ^2^ molecular subtypes based on DEGs.

**Table 2 cancers-14-03582-t002:** Characteristics of patients in GSE54467.

Variables	Number of Cases
***Age*****(*years*)** ≤60/>60	47/32
***Gender*** Male/Female	50/29
***Stage*** I/II/III/not reported	29/29/20/1
***Event*** Alive/Dead	27/52

**Table 3 cancers-14-03582-t003:** Characteristics of patients in IMvigor210.

Variables	Number of Cases
***Event*** Dead/Alive	189/109
***Response*** CR ^1^/PR ^2^/SD ^3^/PD ^4^	25/43/63/167

^1^ CR: complete response; ^2^ PR: partial response; ^3^ SD: stable disease; ^4^ PD: progressive disease.

## Data Availability

All data generated or analyzed during this study are included in this published article and its Appendix A.

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
