# Peer review of "The Multi-Omics Landscape and Clinical Relevance of the Immunological Signature of Phagocytosis Regulators: Implications for Risk Classification and Frontline Therapies in Skin Cutaneous Melanoma"

_cancers, 2022, doi:10.3390/cancers14153582_

Round 1

Reviewer 1 Report

The paper entitled: “The Multi-Omics Landscape and Clinical Relevance of the Immunological Signature of Phagocytosis Regulators: Implications for Risk Classification and Frontline Therapies in Skin Cutaneous Melanoma” by Jiahua Xing et al. is interesting but is confusing and does not allow to understand clearly its focus. In my opinion, also a good revision of English is necessary.

Major comments

There are many figures and huge information and is difficult for the reader to understand all the data. The font in some figures is too small.

I would suggest the authors to reformat their results and some of the graphs. Some of the graphs can be converted to supplementary material.

Author Response

Thank you very much for your professional and constructive comments, which greatly help improve the quality of our manuscript. We have uploaded the response file (please see the attachment) and would like to express our sincere gratitude for your suggestions.

Reviewer 2 Report

Xing et al. have provided a robust analysis of phagocytosis related receptor expression profiles, and validated them using RTPCR and IHC from normal and tumor tissues and in cell lines. I do not have any further questions for this manuscript.

1. What is the main question addressed by the research?

The authors have created for scoring models that utilize expression data of phagocytosis regulators for survival prediction in skin cutaneous melanoma. Their model can be extrapolated to predict success of immunotherapy approaches where therapeutic antibody treatment is expected to activate macrophages to phagocytose and clear tumor cells.

2. Do you consider the topic original or relevant in the field, and if so, why?

This platform for predicting outcome of immunotherapy or anti-cancer treatments based on phagocytosis receptor expression, and combining trancriptomics, metabolomics data appears to be unique and very relevant to the field. As a cellular immunologist I am unable to comment on the robustness of the bioinformatics programming part, but I believe, it will be useful for groups planning to upscale their findings in lab settings to clinical trials.

3. What does it add to the subject area compared with other published material?

The field is aware that phagocytosis regulators have tremendous impact on the cancer immunotherapy approach. There are several research articles on it. This work uses data from such large-scale studies to predict most relevant biomarkers and signaling pathways to target for better cancer prognosis. I did not find any such bioinformatics tool online that uses the expression of phagocytosis regulators to predict treatment outcome.

4. What specific improvements could the authors consider regarding the methodology?

As a cellular immunologist I am unable to comment on the methodology part. However, the validation of their model using immunohistochemistry and gene expression data of their own, adds confidence to their prediction model.

5. Are the conclusions consistent with the evidence and arguments presented and do they address the main question posed?

The authors have addressed the questions that they propose to work on quite well.

6. Are the references appropriate?

Yes.

7. Please include any additional comments on the tables and figures.

The figures using heatmaps are quite small in print, however they should be okay to follow online.

Author Response

(The authors gave the same response as above.)

Reviewer 3 Report

Table 1. How the clusters were defined. Data source for the list in each cluster should be provided.

Table 1,2,3 title: paitients à patients.

What is the source of Table 3. It is unexplained in the text and no reference cited. Googling IMvigor210 showed bladder and non-bladder data not similar to melanoma.

Last paragraph in page 3: suppose this section for data source. The order of the methodology sections and contents below it need to be reorganized. E.g GSEA data should be under section for functional enrichment analysis.

Section 2.4. What is the reference for this PC formula? How this formula will be applied on future new melanoma patients for clinical practice?

Validation using 15 samples only is a limitation, lacking statistical power.

Fig. 3L, is there significantly enriched chromosomal region?

Results of Fig. 7B and 10H are not impressively strong enough.

Fig. 11 IHC are hazy.

Fig. S3. Figures of Atlas of Protein website should be enlarged to visually compare between the normal and abnormal tissues at the cellular levels. So, suggest each of them have a pop out square with the larger resolution.

Table S1. Is of no use in this format. How this is related to the paper goals. Functional enrichment analysis can be analyzed and presented in different ways. Fig. 2 is enough.

Table S2. Suggest (i) adding source of information in the footer of the tables, (ii) keeping significant genes only with cutoff for the correlation coefficient, (iii) type of test used for correlation should be mentioned.

Table S3: what is p_pm stands for and why repeated twice. What p_ stands for. Abbreviations could be explained in footer. What are these values stands for?

Table S4. Contains a lot of columns of no use e.g se and z of the r. This could be converted to nice figure x axis correlation coefficient from -1 to 1 and Y axis gene name and line from 0 extending to r coefficient. Also, cox regression show unuseful columns only HR and its CI is enough and also could be plot, removing insignificant genes. Figure could be like forest plot.

Author Response

(The authors gave the same response as above.)

Round 2

Reviewer 1 Report

The manuscript is improved. 

In this version, the manuscript is accepted for publication. 

Reviewer 3 Report

Comments were addressed.